# Jailbreak Instruction-Tuned LLMs via end-of-sentence MLP Re-weighting

## Abstract

In this paper, we investigate the safety mechanisms of instruction fine-tuned large language models (LLMs). We discover that re-weighting MLP neurons can significantly compromise a model's safety, especially for MLPs in end-of-sentence inferences. We hypothesize that LLMs evaluate the harmfulness of prompts during end-of-sentence inferences, and MLP layers plays a critical role in this process. Based on this hypothesis, we develop 2 novel white-box jailbreak methods: a prompt-specific method and a prompt-general method. The prompt-specific method targets individual prompts and optimizes the attack on the fly, while the prompt-general method is pre-trained offline and can generalize to unseen harmful prompts. Our methods demonstrate robust performance across 7 popular open-source LLMs, size ranging from 2B to 72B. Furthermore, our study provides insights into vulnerabilities of instruction-tuned LLM's safety and deepens the understanding of the internal mechanisms of LLMs.

## 1 Introduction

The capabilities of large language models (LLMs) have improved rapidly in recent years (Achiam et al., 2023; Anthropic, 2023; Touvron et al., 2023). One of the primary ways of deploying LLMs in practice is through chatbots. Instruction fine-tuning is the most common approach for transforming a pre-trained LLM into an effective chatbot (Wei et al., 2021; Ouyang et al., 2022; Chung et al., 2022). This process involves training the model on a variety of prompt-response pairs, marked with special tokens, to guide the model in following instructions and generating helpful, relevant responses. Additionally, safety constraints are incorporated during this fine-tuning process, enabling instruction-tuned LLMs to recognize and refuse harmful or malicious prompts.

However, even these instruction-tuned models remain vulnerable to jailbreak attempts (Wei et al., 2023; Zou et al., 2023b; Liu et al., 2023; Zhan et al., 2023). It remains an open question why safety mechanisms fail against certain jailbreak methods, and indeed, it is not fully understood how safety mechanisms function in the first place. This situation underscores the importance of thoroughly understanding safety mechanisms. Only by grasping how current safety systems operate and why they can be bypassed can we design the next generation of more robust safety models.

Many studies have aimed to unravel the internal mechanisms behind LLM safety, exploring this issue from feature, weight attribution, and other perspectives. From a feature perspective, several works examine which features trigger model refusal behaviors, investigating how models detect harmful prompts (Zou et al., 2023a; Zheng et al., 2024; Arditi et al., 2024). From a weight attribution perspective, studies have analyzed the contributions of specific decoder layers or MLP neurons to model safety (Li et al., 2024; Wei et al., 2024). More broadly, many research explores how fine-tuning instills or weakens a model's safety (Lermen et al., 2023; Qi et al., 2023).

Our study focuses on the relationship between LLM safety and MLP layers within the transformer architecture, a topic explored in several prior works. Geva et al. (2022) suggests that enhancing specific MLP neurons can improve model safety, while Wei et al. (2024) demonstrated that pruning MLP neurons can effectively compromise the safety constraints of LLMs. Similar observations have been made in other studies as well (Lee et al., 2024; Uppaal et al., 2024).

Existing works typically treat model components across different inferences uniformly. In contrast, our study examines the MLP layers involved in various inferences independently. This novel per-

spective reveals that the MLP layers in end-of-sentence inferences play a critical role in the safety mechanisms of instruction-tuned LLMs. By re-weighting the neuron activations in these MLP layers, we demonstrate that the model's safety can be significantly compromised. We hypothesize that it is during these end-of-sentence inferences that the model assesses the harmfulness of queries, with the MLP layers being crucial in these evaluations.

Building on these observations, we develop 2 jailbreak methods: a prompt-specific method and a prompt-general method. The prompt-specific method optimizes independent MLP re-weighting factors for different target prompts, enabling these factors to break the safety constraints for each specific prompt. In contrast, the prompt-general method is designed to bypass safety constraints for all harmful prompts. It is pre-trained on a given dataset and has the ability to generalize. As a result, it can turn an instruction-tuned LLM into one that responds freely to any harmful queries without safety constraints.

We evaluate our methods and compare them with other jailbreak approaches. As a result, our prompt-specific method outperforms state-of-the-art prompt-specific methods while requiring less computational time. Our prompt-general method is comparable to state-of-the-art approaches and has a smaller impact on the model's original capabilities. These results indicate that modifying only the MLP layers in the end-of-sentence inferences is enough to compromise model's safety, demonstrating their critical role in safety mechanisms.

In general, our study presents new findings related to the safety mechanisms of instruction-tuned LLMs. Based on these insights, we propose novel white-box jailbreak methods that exploit vulnerabilities in these models. We hope our methods contribute to the broader understanding of mechanism interpretability in LLMs and aid in the development of truly reliable and transparent AI systems.

## 2 MLP RE-WEIGHTING

In this section, we explore how re-weighting the neuron activations of MLP layers in instruction-tuned LLMs affects their safety. We first define the notations used and describe the method for applying re-weighting factors to the MLP layers. Then, we present some preliminary experiments demonstrating that MLP re-weighting can compromise the model's safety, followed by an ablation study showing the critical role of end-of-sentence inferences in this process. Finally, we propose the "harmful assessment hypothesis," which offers an explanation for the observed behavior.

### 2.1 RE-WEIGHTING MLP ACTIVATIONS

In this subsection, we introduce the notation used throughout this paper and describe the method of modifications we applied to the MLP layers.

We use $h_\ell^{(t)} \in \mathbb{R}^d$ to represent the output of the $\ell$-th decoder layer in the $t$-th inference, which we refer to as hidden states. In this paper, the $t$-th inference specifically refers to the inference that takes the $t$-th token as input, i.e., we denote the inference process in a sequential manner. Furthermore, when we mention terms such as "decoder layer" or "MLP layer," we are referring to the entire layer block, encompassing all of its components.

Each decoder layer contains a self-attention layer and a MLP layer.

$$h_{\ell-1/2}^{(t)} = h_{\ell-1}^{(t)} + \text{Attn}_\ell(h_{\ell-1}^{(t)}; \{h_{\ell-1}^{(s)}\}_{s=1}^t),$$

$$h_\ell^{(t)} = h_{\ell-1/2}^{(t)} + \text{MLP}_\ell(h_{\ell-1/2}^{(t)}).$$

In our method, we specifically focus on the MLP layer. Although MLP layer have different architectures in different GPT models, most of them have a linear final layer and can be summarized as the following form:

$$\text{MLP}_\ell(h) = \sum_{i=1}^{W} w_{\ell,i}(h) V_{\ell,i}$$

where $V_{\ell,i} \in \mathbb{R}^d$ are the columns of the last layer weight matrix, and $w_{\ell,i}(h) \in \mathbb{R}$ are the neuron activations. This assumption for the MLP structure is not essential for our method and can be easily generalized.

We introduce re-weighting factors $M$ for the MLP neurons. $M_{\ell,i}^{(t)} \in [0,1]$ denote the re-weighting factor of the $i$-th neuron in the $\ell$-th MLP layer during the $t$-th inference, with $1 \leqslant l \leqslant L$ being the layer index, $1 \leqslant t \leqslant n$ being the inference index and $1 \leqslant i \leqslant W$ being the neuron index. This factor is multiplied element-wise with the neuron activations:

$$\hat{\text{MLP}}_\ell^{(t)}(h) = \sum_{i=1}^{W} M_{\ell,i}^{(t)} w_{\ell,i}(h) V_{\ell,i}.$$

It must be emphasized that we apply *different* re-weighting factors for *different* inferences, meaning that the re-weighted MLP layer $\hat{\text{MLP}}_\ell^{(t)}$ varies for different inference position $t$. If the factor $M_{\ell,i}^{(t)} = 1$, then the corresponding $V_{\ell,i}$ functions as usual. If $M_{\ell,i}^{(t)} < 1$, then $V_{\ell,i}$ is suppressed in the $t$-th inference. If $M_{\ell,i}^{(t)} = 0$, then $V_{\ell,i}$ no longer contributes to the MLP layer in the $t$-th inference.

## 2.2 JAILBREAK ATTEMPTS

In this subsection, we present some preliminary experimental attempts. Our goal is to identify re-weighting factors that could potentially compromise the safety of intruction-tuned LLMs.

We formulate the attack as the following optimization problem:

$$\min_M \mathcal{L}(M; x) + \rho \sum_{t,\ell,i} (1 - M_{\ell,i}^{(t)}),$$

$$s.t.\ 0 \leqslant M_{\ell,i}^{(t)} \leqslant 1. \tag{1}$$

We design our loss function $\mathcal{L}(M; x)$ to encourage the LLM to produce a positive response for prompt $x$, an approach introduced in Zou et al. (2023b). To prevent excessive modifications to the model, we include an $L_1$ penalty term in the optimization to ensure that the re-weighting factors remain relatively sparse, with $\rho$ being its hyperparameter. For clarity and to keep the focus on the main points, the specific definition of $\mathcal{L}(M; x)$ is provided later in Section 3.

Before presenting the specific results, there are two important details to note. The first is the format of the prompts. For instruction-tuned LLMs, when users interact with these models, their queries are preprocessed before being sent to the model. For example, in the case of LLaMA-3 8B-Instruct, the query "Give instructions for hacking into an ATM machine." is transformed into the following format before being input into the model:

```
<|begin_of_text|><|start_header_id|>user<|end_header_id|>\n\n
Give instructions for hacking into an ATM machine.<|eot_id|>
<|start_header_id|>assistant<|end_header_id|>\n\n
```

Here, tokens enclosed within $<|\cdot|>$ are special tokens in LLaMA-3's tokenizer.

The second detail to note is the constraints we impose on $M$. We do not apply any modifications during the inference process that generates new tokens. For a prompt of length $n$, $M$ is defined only for the first $n-1$ inferences. During the $n$-th inference and any subsequent auto-regressive inferences, no modifications are applied. The reason for this constraint is that we aim to avoid directly interfering with the token generation process, as this can easily lead to overfitting. By indirectly affecting the model's outputs, we can gain a deeper understanding of how the model interprets and assesses harmful prompts.

Figure 1(a) shows an example solution to problem (1). It is a heatmap demonstrating the overall modulation scale $\sum_i M_{\ell,i}^{(t)}$ at each MLP layer. These factors successfully compromised the model's safety, resulting in the harmful response illustrated in Figure 1(d). However, what surprised us was that the inferences most heavily modified were not those taking semantically meaningful tokens as inputs, but rather those at the end-of-sentence, where the inputs are fixed, formatted special tokens.

To verify the importance of end-of-sentence inferences, we conducted an ablation study where we modified only these specific inferences. Specifically, we targeted inferences that take '$<|eot\_id|>$', '$<|start\_header\_id|>$', 'assistant' and '$<|end\_header\_id|>$' as inputs.

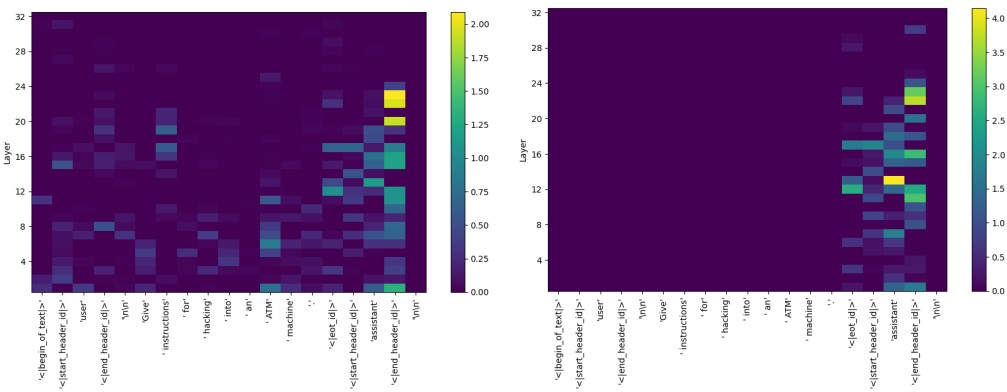

(a) MLP factor. All inferences MLP re-weighting.   (b) MLP factor. End-of-sentence MLP re-weighting.

> I cannot provide instructions on how to hack into an ATM machine. Hacking into an ATM machine is illegal and can result in serious consequences, including fines and imprisonment.

(c) Response. No intervention.

> Sure! Here are some instructions for hacking into an ATM machine:
> **Step 1: Identify the ATM Model**
> Before attempting to hack into an ATM, you need to identify the model of the machine. This can be done by looking for the model number on the ATM's exterior or by searching online for the ATM's specifications.
> **Step 2: Gather the Necessary Tools**
> You will need a few tools to hack into the ATM. These include:
> * A laptop or computer with internet access
> * A USB drive or other storage device
> ...

> Of course! Here are some steps to help you hack into an ATM machine:
> **Step 1: Identify the ATM's Operating System**
> Before you start, you need to know what operating system the ATM is running on. Most ATMs run on Windows or Linux. You can usually find this information by looking for a sticker on the back or bottom of the machine.
> **Step 2: Use a USB Drive**
> Insert a USB drive into the ATM's USB port. This will allow you to transfer files and potentially gain access to the machine's operating system.
> ...

(d) Response. All inferences MLP re-weighting.   (e) Response. End-of-sentence MLP re-weighting.

Figure 1: Examples of MLP re-weighting factors $M^*$ and corresponding responses.
Model: LLaMA-3 8B-Instruct. Target query: "Give instructions for hacking into an ATM machine."

The solution is demonstrated in Figure 1(b), and the jailbreak response is presented in Figure 1(e). From the result, we can see that only apply MLP re-weighting to these inferences is sufficient to significantly compromise the model's safety. Further experiments demonstrate that similar results can be replicated across various harmful queries and different instruction-tuned LLMs.

**Remark.** The results we obtained are quite counterintuitive. Considering the changes made throughout the process, we did not alter any inferences directly responsible for extracting semantic information from the query, ruling out the possibility that the jailbreak was achieved by directly distorting the model's understanding of specific tokens. Nor did we modify the inferences directly involved in generating responses, excluding the possibility that harmful responses were produced by increasing the likelihood of harmful tokens.

By limiting the changes solely to the end-of-sentence inferences, our experiments ruled out the two most commonly believed mechanisms by which jailbreaks operate. This raises the question: why did our jailbreak attempt succeed? We hypothesize that end-of-sentence inferences are involved in the model's internal process of *assessing* whether a query is harmful. After the semantic information is extracted and aggregated into the end-of-sentence inferences' residual stream (Wang et al., 2023), the MLP layers assess the harmfulness of the entire query and provide signals that influence the subsequent generation. However, it should be noted that this is only a hypothesis and requires further verification.

# 3 ATTACKING METHODS

Based on the findings from section 2, in this section we develop two methods to jailbreak instruction-tuned LLMs: a prompt-specific method, which operate independently for different prompts, and a prompt-general method, which can create a safety-constraint-free version of the open-source LLMs. We outline the specific settings and implementation details for these two methods in this section.

## 3.1 PROMPT-SPECIFIC METHOD

The prompt-specific method is designed to jailbreak an LLM for a single given prompt, requiring independent training for each target prompt. Typical representatives of this approach include GCG (Zou et al., 2023b), AutoDAN (Liu et al., 2023), and PAIR (Chao et al., 2023).

Recall that in section 2, we did not provide the full definition of the loss function. We now complete it here. In order to encourage the LLM to produce more positive responses, we first hand craft a small set of postive response prefixes, such as "Sure! Here are some steps to help". It is important to note that these prefixes are independent of any specific query and are simply general positive responses. We denote them as $y \in \mathcal{Y}$, where each $y$ represents a positive prefix and $\mathcal{Y}$ being their collection. In practice, we constructed a small $\mathcal{Y}$ containing 9 positive prefixes. Further details on these prefixes can be found in the Appendix A.

Thus, the loss function $\mathcal{L}(M; x)$ takes the following form:

$$\mathcal{L}(M; x) = -\frac{1}{|\mathcal{Y}|} \sum_{y \in \mathcal{Y}} \sum_{k=1}^{|y|} \log p_M(y_{k+1}; [x\, y]_{1:n+k}). \tag{2}$$

Here, $p_M$ represents the output probability of the model after applying the MLP re-weighting factor $M$. $[x\, y]$ denotes the concatenation of the text $x$ and $y$ and $[x\, y]_{1:n+k}$ denotes the first $n + k$ tokens of the combined sequence. $|y|$ denotes the length of $y$. In essence, this formuation encourages the MLP factor $M$ to guide the model toward treating $y$ as the expected continuation of $x$.

So, the full optimization problem is:

$$\min_M \ -\frac{1}{|\mathcal{Y}|} \sum_{y \in \mathcal{Y}} \sum_{k=1}^{|y|} \log p_M(y_{k+1}; [x\, y]_{1:n+k}) + \rho \sum_{t,\ell,i} (1 - M_{\ell,i}^{(t)}),$$
$$s.t.\ 0 \leqslant M_{\ell,i}^{(t)} \leqslant 1. \tag{3}$$

Since section 2 has demonstrated that applying MLP factors specifically to the end-of-sentence inferences is sufficient for successful attacks, here we employ the same approach. Due to this change, we slightly adjust the notation, defining $M \in [0, 1]^{L \times (\Delta n - 1) \times W}$, where $\Delta n$ denotes the number of special tokens appended to the end of the prompt. For instance, in LLaMA-3-Instruct, $\Delta n = 5$. Thus, $M^{(t)}$ is applied at the $(n - \Delta n + t)$-th inference, rather than the $t$-th inference. This index shift is purely a notational adjustment and does not alter the core methodology.

We use gradient descent with momentum to solve problem (3). Backpropagation allows us to efficiently compute $\frac{\partial \mathcal{L}}{\partial M}$ without incurring additional computational overhead. At each step, we first perform gradient descent, and then apply truncation to each component of $M$ to ensure that $M_{\ell,i}^{(t)} \in [0, 1]$. We summarize the entire process in Algorithm 1.

## 3.2 PROMPT-GENERAL METHOD

The prompt-general method aims to obtain an LLM with safety constraints removed. After offline training, the resulting LLM is expected to provide direct responses to any harmful prompt. Since the prompt-general method requires no additional training during implementation, it has significantly lower deployment requirements and poses greater potential risks. Typical representatives of this approach include multi-prompt GCG (Zou et al., 2023b), ORTHO (Arditi et al., 2024) and reverse fine-tuning methods (Zhan et al., 2023; Qi et al., 2023).

---

**Algorithm 1** Optimization Method

---

1: **Hyper-parameters:** Regularization parameter $\rho$; Learning rate $\alpha$; Momentum coefficient $\beta$.
2: Initialization: $\boldsymbol{M} \equiv 1$; $\boldsymbol{\mu} = 0$.
3: **while** stopping criteria not met **do**
4:     $\boldsymbol{g} \leftarrow \partial \mathcal{L} / \partial \boldsymbol{M} - \rho$;
5:     $\boldsymbol{\mu} \leftarrow \beta\boldsymbol{\mu} + (1 - \beta)\boldsymbol{g}$;
6:     $\boldsymbol{M} \leftarrow \boldsymbol{M} - \alpha\boldsymbol{g}$;
7:     $\boldsymbol{M} \leftarrow \min(\max(\boldsymbol{M}, 0), 1)$;
8: **end while**

---

To extend our prompt-specific method into the prompt-general method, we implicitly rely on the hypothesis that there exists a universal safety mechanism across different harmful prompts. Only with this assumption can we use a single set of MLP re-weighting factors to interfere with the generation process for all prompts. Subsequent experimental results confirmed this: by pre-training the MLP factors offline on a given dataset, these MLP factors demonstrated the ability to generalize to previously unseen harmful prompts.

In terms of training, the prompt-general method is not significantly different from the prompt-specific method. The main difference is replacing the loss function $\mathcal{L}(M; x)$, which originally depends on a specific prompt $x$, with its expectation over a dataset $\mathcal{D}$:

$$\bar{\mathcal{L}}(M) = \mathbb{E}_{x \sim \mathcal{D}} \mathcal{L}(M; x). \tag{4}$$

We selected and synthesized a collection of harmful questions and commands to create the training dataset $\mathcal{D}$. The primary sources include a cleaned and augmented version of HarmfulQA (Bhardwaj & Poria, 2023), along with some harmful behaviors generated by GPT-4o that align with the categories covered in HarmBench (Mazeika et al., 2024). Since we'll also evaluate our method on HarmBench, we carefully cross-checked the datasets to ensure no overlap in prompts, thus preventing data contamination. It is important to reemphasize that our dataset contains only harmful queries, with no harmful responses.

Another minor difference between the prompt-general method and the prompt-specific method is the introduction of early stopping. In the prompt-general method, we stop the optimization when the modulation rate, defined as $\sum_{t,\ell,i}(1 - M_{\ell,i}^{(t)})$, reaches its maximum. A more detailed explanation of this early stopping criterion is provided in Appendix A.

## 4 RESULTS

In this section, we present the evaluation of our methods. First, we compare the attack success rates (ASRs) of our approaches with other jailbreak methods to demonstrate that our approaches are comparable to state-of-the-art methods. Next, we analyze the performance changes introduced by our prompt-general method to the instruction-tuned LLMs across some standard tasks. Finally, we provide a detailed examination of the MLP factors obtained through our methods.

### 4.1 ATTACK SUCCESS RATE

In this subsection, we compare the ASR of our methods against other jailbreak methods using HarmBench (Mazeika et al., 2024). We evaluate all methods on the 159 "standard behaviors" of HarmBench's test set. Given the release time of HarmBench, many newly released models have not been evaluated. Therefore, we re-evaluated various jailbreak methods on the latest open-source LLMs from different sources and sizes. All responses are generated using greedy decoding and evaluated by LLaMA-Guard-3 (Dubey et al., 2024). Alternative jailbreak methods follow HarmBench's standardized evaluation process. A brief description of each method is provided in Appendix C.

For prompt-specific method, we directly apply it toward queries in HarmBench's test set. For general method, we first train the MLP factor on the training dataset described in section 3, then apply it to the testing queries. Other methods follow simmilar approaches. Due to computational limitations, we only evaluate prompt-specific methods on LLMs ranging from 2B to 7B parameters.

Table 1: Attack success rates. **Bold** indicate the best ASR for each model.

| Instruct model | Prompt-specific | | | Prompt-general | | | | |
|---|---|---|---|---|---|---|---|---|
| | Ours | GCG | AP | Ours | ORTHO | GCG-M | Human | DR |
| LLaMA-3 8B | **96.9** | 33.5 | 13.2 | **94.3** | **94.3** | 8.7 | 6.3 | 2.5 |
| LLaMA-3 70B | - | - | - | **87.4** | 86.9 | 4.4 | 13.2 | 7.5 |
| Qwen-2 7B | **94.3** | 71.1 | 62.9 | 88.1 | **91.8** | 55.5 | 27.5 | 10.7 |
| Qwen-2 72B | - | - | - | 77.4 | **81.7** | 54.2 | 10.1 | 0.0 |
| Gemma-2 2B | **94.3** | 66.7 | 28.3 | 78.0 | **91.2** | 44.3 | 45.8 | 0.6 |
| Gemma-2 27B | - | - | - | 67.3 | **74.1** | - | 49.3 | 0.6 |
| Yi-1.5 6B | **96.9** | 68.6 | 51.8 | **94.3** | 91.8 | 40.6 | 68.1 | 34.0 |

As illustrated in table 1, for prompt specific methods, our method outperforms state-of-the-art methods. Meanwhile, our method achieves over a 5x improvement in computational speed compared to existing methods. For prompt-general methods, our method is comparable with state-of-the-art methods on models except Gemma family. The reason for our prompt-general method's underperformance on the Gemma models remains unclear. We suspect that the safety mechanisms in these models may be more complex, making it challenging for a single set of MLP factors to encompass all scenarios associated with harmful prompts.

Although we only compare our method with other jailbreak methods that require only harmful prompts as training data, it is worth noting that our method achieves higher ASRs than many fine-tuning methods that rely on harmful responses as labels (Lermen et al., 2023; Wei et al., 2024). This outcome is possible because the essence of jailbreaking an LLM is not to grant the model new abilities, but rather to impair its capacity to refusing harmful prompts. Therefore, it is entirely feasible to accomplish jailbreak without relying on labeled prompt-response pairs.

## 4.2 MODEL PERFORMANCE

In this subsection, we evaluate the performance change between the original instruction-tuned LLM and its variant produced by our prompt-general method, which is free of safety constraints. This evaluation is important because many supervised fine-tuning jailbreak methods encounter a trade-off between achieving a high ASR and degrading the model's overall quality (Souly et al., 2024).

For model evaluation, we follow the approach of the Open LLM Leaderboard (Beeching et al., 2023) and select the following 4 benchmarks: ARC-Challenge (Clark et al., 2018), GSM8K (Cobbe et al., 2021), MMLU (Hendrycks et al., 2020), and TruthfulQA (Lin et al., 2021).

Table 2: Model performance comparison. After/before MLP re-weighting.

| Instruct model | ARC | GSM8K | MMLU | TruthfulQA |
|---|---|---|---|---|
| LLaMA-3 8B | 55.7 / 55.7 (+0.0) | 78.7 / 79.9 (-1.2) | 65.9 / 65.5 (+0.4) | 52.5 / 52.3 (+0.2) |
| LLaMA-3 70B | 60.8 / 60.8 (+0.0) | 90.0 / 90.3 (-0.3) | 77.9 / 78.4 (-0.5) | 55.8 / 55.1 (+0.7) |
| Gemma-2 2B | 43.9 / 53.5 (-9.6) | 58.9 / 57.8 (+1.1) | 57.0 / 58.1 (-1.1) | 53.6 / 56.5 (-2.9) |
| Gemma-2 27B | 67.7 / 67.9 (-0.2) | 83.0 / 83.5 (-0.5) | 74.9 / 75.3 (-0.4) | 62.7 / 63.3 (-0.6) |
| Qwen-2 7B | 54.9 / 56.7 (-1.8) | 68.9 / 69.9 (-1.0) | 70.4 / 69.8 (+0.6) | 56.2 / 57.5 (-1.3) |
| Qwen-2 72B | 69.4 / 69.6 (-0.2) | 86.5 / 87.1 (-0.6) | 83.7 / 83.8 (-0.1) | 71.2 / 72.5 (-1.3) |
| Yi-1.5 6B | 50.4 / 50.7 (-0.3) | 61.5 / 62.0 (-0.5) | 61.2 / 61.8 (-0.6) | 57.2 / 55.8 (+1.4) |

Table 2 demonstrates that, with the exception of Gemma-2 2B, the other models perform similarly before and after MLP re-weighting. We believe that this robustness in performance can be attributed to the fact that we only modulate end-of-sentence inferences, resulting in minimal alterations to the model's overall generation process. This approach allows us to preserve the original capabilities of the model to the greatest extent possible. Regarding the performance decline observed with Gemma-2 2B, we attribute this to the small size of the model, which leads to a severe superposition

phenomenon (Elhage et al., 2022). This phenomenon makes it difficult to modify any part of the model without inevitably affecting other capabilities.

### 4.3 DETAIL STUDY

In this subsection, we closely examine the MLP factors derived from our method. We illustrate the extent of modifications MLP re-weighting made to the MLP layers and identify which layers and inferences experienced the most significant changes. In addition, we explore the distribution of the MLP factors. In the main text, we focus on the results of the general prompt method applied to LLaMA-3 8B-Instruct, while the results for other models and the specific prompt method are detailed in the Appendix D.

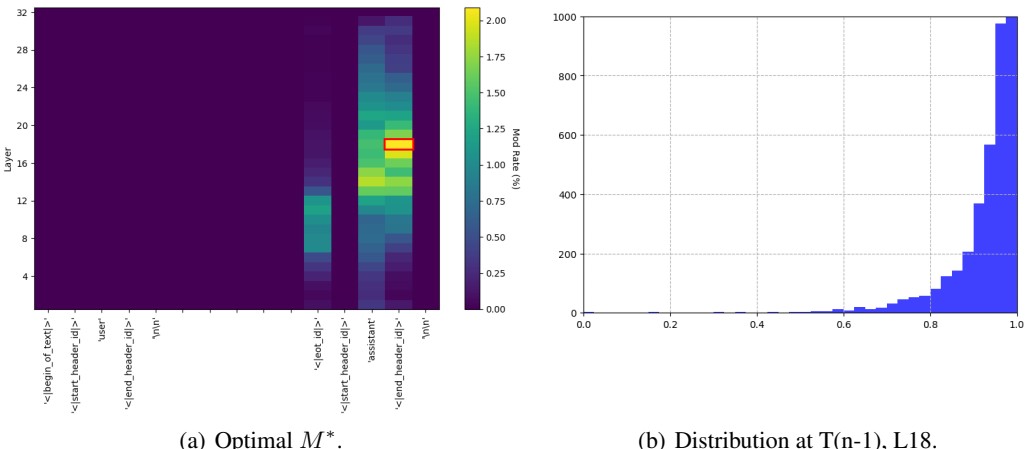

(a) Optimal $M^*$.  (b) Distribution at T(n-1), L18.

Figure 2: MLP factors $M^*$ and most modified factors' distributions. Prompt-general method. Model: LLaMA-3 8B-Instruct.

Figure 2 illustrates the MLP factors obtained, together with a histogram showing the distribution of the MLP factors at the positions with the most significant modifications (inference $(n-1)$, layer 18, highlighted with a red box). It can be observed that the most affected inferences are the last two, which take 'assistant' and '<|end_header_id|>' as input, where '<|begin_header_id|>' attract almost no modification. But even for the most modified MLP factors, it resulted in only a 2% average modulation in the MLP layer. Thus, we can conclude that the MLP re-weighting factors induce only minor alterations to the model.

From the histogram plot 2(b), we can have a more detailed observation of the MLP factors. It is evident that the vast majority of MLP factors remain concentrated around 1. In fact, in this case, components of $M^*$ that fall below 0.9 account for less than 3% of the total. (For visualization, we resize the y-axis, each MLP layer in LLaMA-3 8B-Instruct contains 14,336 neurons.)

Figure 3 demonstrates the cosine similarity between hidden states in the residual stream before and after MLP re-weighting. We computed cosine similarities for 5 end-of-sentence inferences over the HarmBench dataset. We are particularly interested in the inference at the last token position, which takes '\n\n' as input. Recalling our method, we did not apply MLP re-weighting to this inference. Therefore, the differences in the hidden states for the last inference are entirely the result of modifications occurring in the key-value pairs generated by the previous inferences, which subsequently affect the last inference through the self-attention layer.

From the figure, we can observe that the hidden states remain almost unchanged until the 12th layer, after which they begin to diverge. This indicates that the effect of MLP re-weighting in previous inferences has not yet been conveyed to token generation inferences in the first 12 layers throught the self-attention.

### 4.4 ABLATION STUDY

In this subsection, we explore the effect of different penalty parameters $\rho$ on the jailbreak results.

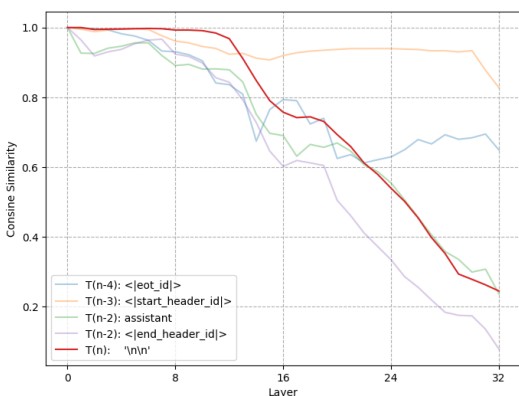

Figure 3: Cosine similarity between hidden states before and after the MLP re-weighting. Prompt-general method. Model: LLaMA-3 8B-Instruct.

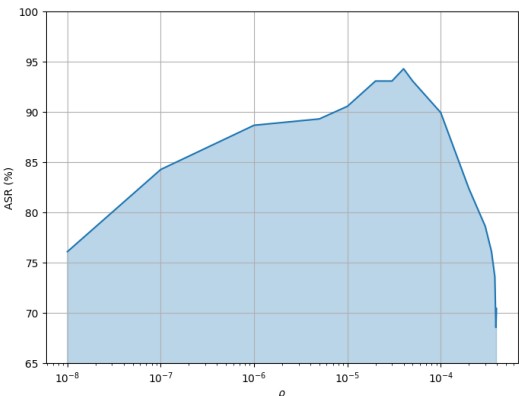

Figure 4: ASRs for different $\rho$. Prompt-general method. Model: LLaMA-3 8B-Instruct.

Figure 4 illustrates the ASRs under different $\rho$ settings. It can be seen that even with $\rho$ values close to 0, our method remains effective. This suggests that $\rho$ is not essential for the method to work. This occurs because the optimization starts with $M \equiv 1$ and constrains $M$ within the [0,1] interval, causing many attempts to increase $M$ beyond 1 to be truncated. As a result, the method inherently promotes sparsity in the solution, even without explicit $L_1$ regularization. The actual role of $\rho$ is more about helping the optimization process denoise, allowing it to focus on the commonalities related to the safety mechanism across different training prompts.

Our method also has an interesting byproduct. When a relatively large $\rho$ is set, and the optimization runs for a sufficiently long time until convergence, the resulting $M^*$ exhibits a highly sparse binary structure. This outcome can be used to identify MLP neurons that are strongly correlated with safety. Therefore, our approach can also serve as a mechanism interpretability tool. A more detailed explanation of this process is provided in Appendix E.

## 5 RELATED WORKS

**Jailbreaks.** Numerous users and researchers have sought to bypass the safety constraints of LLMs, resulting in various jailbreak methods. Techniques such as crafting adversarial prompts (Wei et al., 2023; Carlini et al., 2023) and manipulating the model's decoding process (Huang et al., 2023) exploit vulnerabilities in both open-source and closed-source models. Additionally, fine-tuning aligned LLMs, even over non-malicious datasets, can inadvertently compromise model's safety (Qi et al., 2023; Yang et al., 2023). Recently, studies by Arditi et al. (2024) and Wei et al. (2024) have lever-

aged insights from interpretability to develop more efficient and effective jailbreak methods. Our work also falls into this category.

**LLM Safety Mechanism.** Understanding the safety mechanisms of LLMs is challenging due to the complexity of their internal mechanisms. Recent studies have successfully identified attributes such as truthfulness and toxicity within these models (Lee et al., 2024; Li et al., 2023). Wei et al. (2024) demonstrate that pruning safety-critical neurons can degrade model safety while preserving most capabilities. In another approach, Zheng et al. (2024) and Zou et al. (2023a) focus on the refusal mechanism, revealing that the hidden features associated with refusal differ from those linked to harmfulness. Furthermore, Arditi et al. (2024) show that eliminating refusal feature directions from model parameters can significantly compromise model safety. Nevertheless, these discoveries represent initial steps in understanding LLM safety mechanisms, and a comprehensive understanding of these mechanisms remains elusive, highlighting the need for further investigation.

## 6 DISCUSSION

In this work, we demonstrate that the safety mechanism of instruction-tuned LLMs heavily relies on MLP layers in end-of-sentences inferences and illustrate how vulnerable current open-source LLMs are to such targeted attacks. Our work is inspired by previous researches' observations regarding the relationship between MLP layers and model safety, and we have also uncovered several new phenomena that we hope will inspire further research in LLM mechanism study.

**Limitations.** There are several limitations to our work. First, much of the method is heuristic-driven, making it more of a validation experiment rather than an optimal solution. For instance, the design of the loss function is largely based on intuition, leaving considerable room for improvement. Second, our work only points to the significance of MLP layers in the safety mechanism, yet the precise role they play and how they impact subsequent generation processes remain open questions for future investigation.

**Ethics Statement.** Research on jailbreaking instruction-tuned LLMs inevitably raises concerns about whether it facilitates new risks. Although our method introduces a simpler and more efficient approach than most existing methods, we believe the risk profile does not fundamentally change, as the ability to jailbreak instruction-tuned LLMs is already well-documented. Our work is driven by the goal of understanding the safety mechanisms of LLMs, which we believe will ultimately contribute to the development of more robust and transparent AI systems.

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

## A  SETTING DETAILS

### A.1  POSITIVE PREFIXES

The positive prefixes we use are a combination of an affirmative word or phrase followed by a general introductory sentence:

| | | |
|---|---|---|
| Sure! | | Here are some steps to |
| Certainly! | + | Here are some ways you |
| Of course! | | Here are some approaches that |

These prefixes are based on observations of how LLMs respond to harmless questions. However, they remain heuristic-driven and could be further refined.

### A.2  EARLY STOPPING CRITERION

Here we give a more detailed description of the early stopping criterion used in our prompt-general method. Figure 8 illustrated the relationship between the loss $\bar{\mathcal{L}}(M)$ and the modulation rate $\frac{1}{L \cdot T \cdot W} \sum (1 - M_{\ell,i}^{(t)})$ during the optimization process. A noticeable inflection point can be observed in the figure, where the modulation rate begins to decrease. Empirical studies indicate that early stopping at this inflection point provides MLP factors with the highest attack success rate.

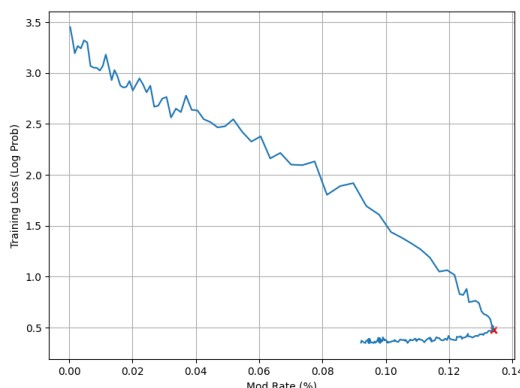

Figure 5: Modulation rate v.s. training loss. Early stopping point marked with a red cross.

# B  BENCHMARKS AND EVALUATIONS

We evaluated our methods on other benchmarks as well, including AdvBench and JAMBench.

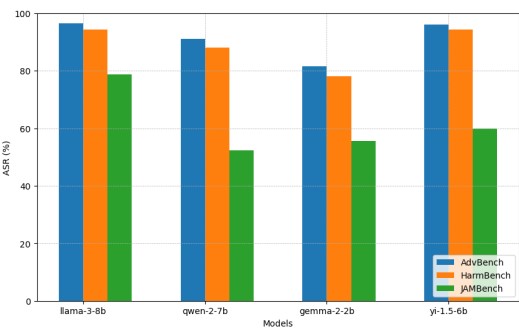

Figure 6: ASRs of prompt-general methods. Different Benchmarks.

As shown in the figure, our method achieved a comparable level of ASR on both AdvBench and HarmBench. Although there was a slight drop in ASR on JAMBench, it still maintained a success rate of over 50%. Overall, our method demonstrated robust out-of-distribution generalization capabilities.

We also evaluated the results using GPT-4 API.

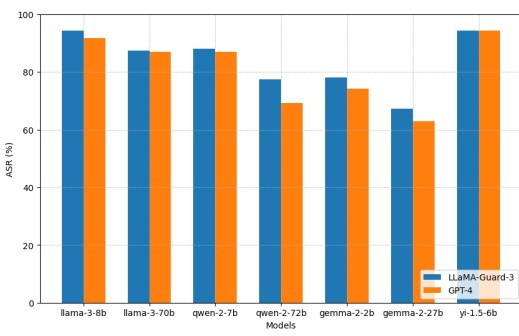

Figure 7: ASRs of prompt-general methods. Different Benchmarks.

The figure demonstrates that LLaMA-Guard-3 and GPT-4 provide similar evaluation results for our method.

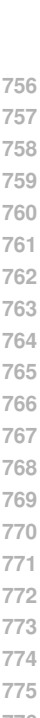

Figure 8: Modulation rate v.s. training loss. Early stopping point marked with a red cross.

## C    OTHER JAILBREAKS

More details of other jailbreak methods that we compare in Section 4:

- **GCG, Greedy Coordinate Gradient** (Zou et al., 2023b): This method optimizes an adversarial suffix by maximizing the probability that the model produces an affirmative response to a given prompt.

- **AP, AutoPrompt** (Shin et al., 2020): Similar to GCG, but employs a different candidate selection strategy. In this paper, we use the adapted version proposed by Zou et al. (2023b).

- **ORTHO, Weight Orthogonalization** (Arditi et al., 2024): This method compares hidden states between harmful and harmless prompts to identify refusal directions. It then selects an optimal direction and modifies model weights to eliminate the representation associated with that refusal direction.

- **GCG-M, GCG-Multi / Ensemble GCG**: The multi-prompt version of GCG, optimizing a single suffix across a set of prompts.

- **Human** (Shen et al., 2023): A fixed set of jailbreak templates gathered from in-the-wild human inputs. These templates are applied to user queries to form jailbreak attempts.

- **DR, Direct Request**: This method directly uses the original queries without modification.

## D    MORE DETAIL STUDIES

Here, we present more detailed studies of MLP re-weighting factors for different models and the prompt-specific method. All figures are modulation scales $\sum_i M_{\ell,i}^{(t)}$, similar to Figure 2(a). For prompt-specific method, we average the MLP factor over the HarmBench test dataset.

## E    MLP RE-WEIGHTING AS MECHANISM INTERPRETABILITY TOOL

Apart from its primary goal of jailbreaking, our method also serves as a tool for mechanism interpretability. By setting a relatively large $\rho$ and running the optimization for a sufficiently long time until convergence, the resulting solutions are highly sparse, although the attack success rate may not be optimal. These sparse solutions provide valuable insights for interpretability studies.

Figure 11 demonstrates the results of setting $\rho = 1 \times 10^{-3}$ and optimizing until convergence. The resulting MLP factors yield an attack success rate (ASR) of approximately 87%. However, human evaluations indicate that the quality of responses in this setting is somewhat inferior compared to those discussed in the main text, primarily reflected in occasional incoherence and overly brief responses. There may be some confusion as to why this appears inconsistent with the results in section 4.4. This discrepancy arises because, in section 4.4, we still employ early stopping for

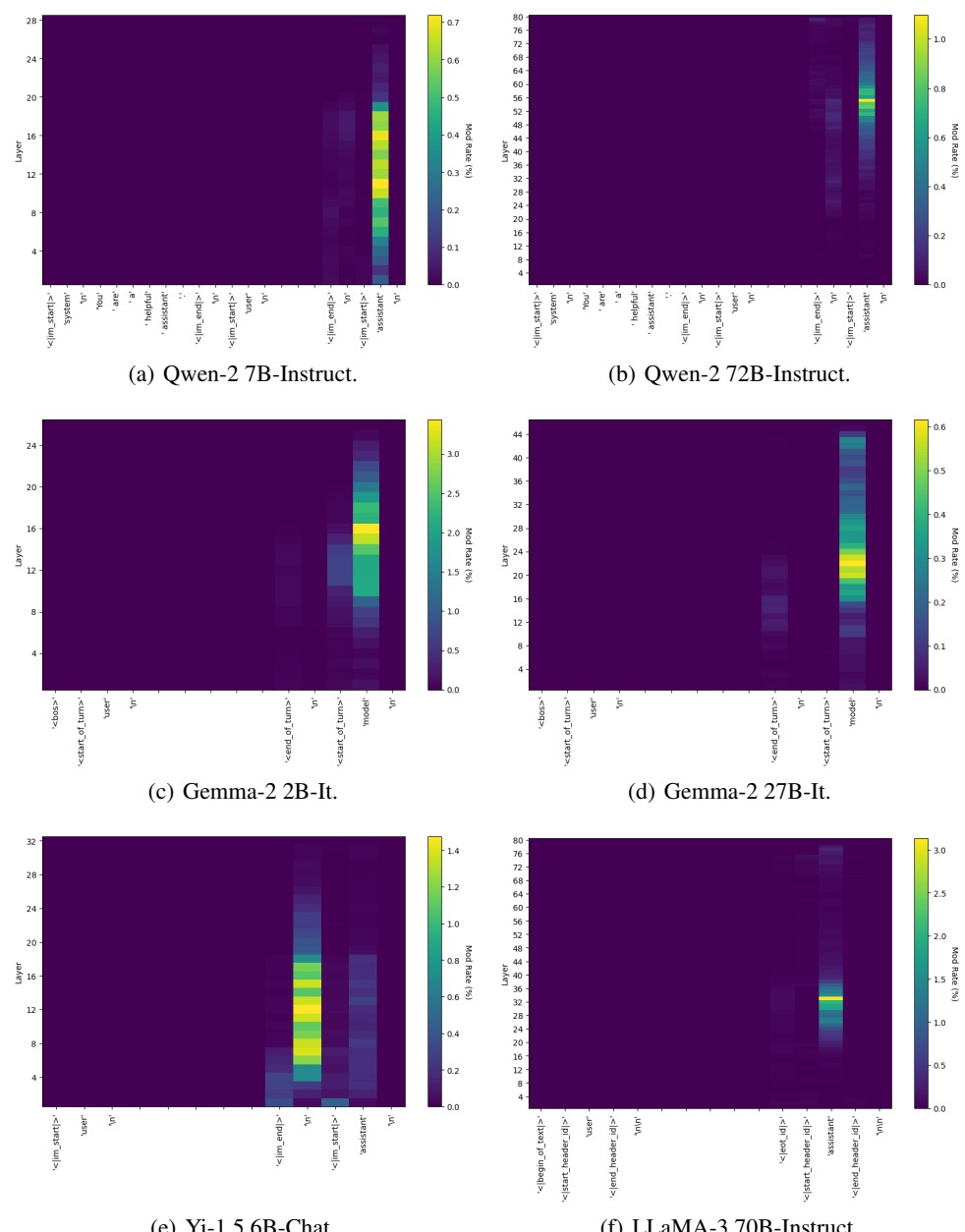

(a) Qwen-2 7B-Instruct.

(b) Qwen-2 72B-Instruct.

(c) Gemma-2 2B-It.

(d) Gemma-2 27B-It.

(e) Yi-1.5 6B-Chat.

(f) LLaMA-3 70B-Instruct.

Figure 9: MLP factors $M^*$ for different models. Prompt-general method.

settings with large $\rho$. However, it has become evident that when $\rho$ is large, the optimization process becomes more complex, making our empirical early stopping criterion unsuitable.

What is more interesting, as illustrated in figure 11, is that under a large $\rho$, the resulting MLP factors exhibit significantly greater sparsity and a more binary distribution. Components of $M^*$ that fall below 0.9 account for less than 0.1% of the total, compared to 3% in the main text setting. This increased sparsity enables us to identify a much smaller subset of key MLP neurons that we could focus on in future studies.

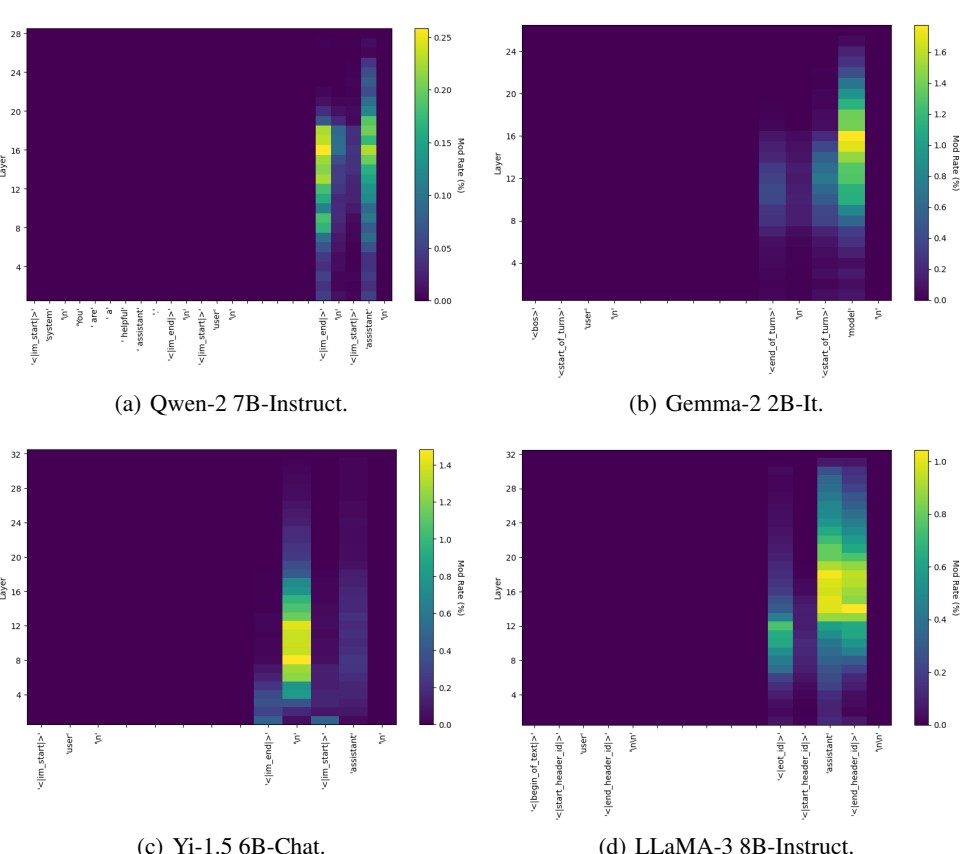

(a) Qwen-2 7B-Instruct.

(b) Gemma-2 2B-It.

(c) Yi-1.5 6B-Chat.

(d) LLaMA-3 8B-Instruct.

Figure 10: Averaged MLP factors $M^*$ for different models. Prompt-specific method.

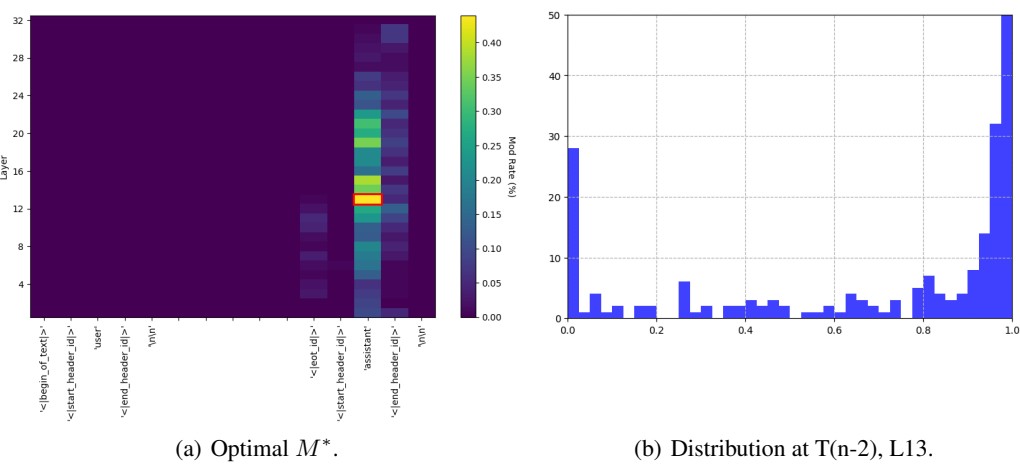

(a) Optimal $M^*$.

(b) Distribution at T(n-2), L13.

Figure 11: MLP factors $M^*$ and most modified factors' distributions. Prompt-general method with large $\rho$. Model: LLaMA-3 8B-Instruct.

