# OpenReview forum: "Jailbreak Instruction-Tuned Large Language Models via MLP Re-weighting"
_ICLR.cc/2025/Conference — ICLR 2025 Conference Withdrawn Submission_

### Official Review · Reviewer_NJHF · 2024-11-02

**Soundness:** 2
**Presentation:** 2
**Contribution:** 1
**Rating:** 3
**Confidence:** 5

**Summary:**

In short, this paper is modifying the GCG attack by only optimizing the MLP layer. The work is incremental. The observation of high-attention score on special tokens seems to be highly related to the attention sink paper, and I would recommend author to reconsider the explanation provided in the paper.

**Strengths:**

The research problem in the paper is important.
The paper made an attempt to explain the interesting phenomenon of high attention scores on special tokens.

**Weaknesses:**

See the questions section.
Overall, I barely learn anything new from the paper. Jailbreaking prompts can break the model and the llm attention sink phenomenon are well known to the community. I highly recommend the author focus on exploring new threat models in new production models or modalities.

**Questions:**

1. Can you explain what does this phrase mean? L13 "especially for MLPs in end-of-sentence inferences"
2. After reading this sentence, "This situation underscores the importance of thoroughly understanding safety mechanisms.". I thought the author is going to propose a defense, but instead, the paper only focuses on the attack.
3. Suggest to add citations for "Many studies have aimed to unravel the internal mechanisms behind LLM safety, exploring this issue from feature, weight attribution, and other perspectives.".
4. Why study MLP? what's the motivation? And why is it a novel perspective, given the authors have cited many previous studies on this topic?
5. Define "end-of-sentence" inference
6. Is reweighting the same as randomly perturbing the model weights? If so, it's not surprising that reweighting lowers the performance, which in this paper's context, undermining the model's safety.
7. L66 "We evaluate our methods and compare them with other jailbreak approaches". Specify the name explicitly
8. L66, " As a result" result of what?
9. L69, "has a smaller impact on the model" how small?
10. L73, "presents new findings", " Based on these insights", what findings and what insights? can you list them as bullets?
11. L114, the reweighting sounds like another dropout layer?
12. L140, suggest to move prompt template details to appendix
13. How is the proposed idea different from the GCG attack? It reads like a GCG optimized on MLP layer
14. L158, "but rather those at the end-of-sentence, where the inputs are fixed, formatted special tokens", this may be related to LLM attention sink paper (Efficient Streaming Language Models with Attention Sinks). Fig. 1 again confirms my guess on attention sink: the first few tokens of the generation has high attention score.
15. Tbh, this is not end-of-sentence, it should be the beginning of the model generation.
16. L222. Drop this sentence "We now complete it here. "
17. How do you differentiate your attack from existing jailbreaking attacks?
18. Missing a summarization figure that introduces the main paper idea
19 The writing needs significant improvements. The definitions are jumping around, and the some sentences are too informal to appear in a paper.

---

> ### Author Response · Authors · 2024-11-15
>
> > The research problem in the paper is important. The paper made an attempt to explain the interesting phenomenon of high attention scores on special tokens.
>
> Our work has nothing to do with attention scores. In fact, none of the results presented in the paper are related to attention scores. While there may be internal connections between these phenomena, exploring them is beyond the scope of this paper.
>
> > Can you explain what does this phrase mean? L13 "especially for MLPs in end-of-sentence inferences"
> >
> > Define "end-of-sentence" inference
> >
> > Tbh, this is not end-of-sentence, it should be the beginning of the model generation.
>
> It’s great to see that the reviewer addressed their own question in the following comments. We chose “end-of-sentence” because it serves as a reminder of the special tokens that appear at the end of a prompt. This aligns with the "eos token" used in many transformer tokenizers, which is why we chose this name.
>
> > Why study MLP? what's the motivation? And why is it a novel perspective, given the authors have cited many previous studies on this topic?
>
> Previous works have demonstrated MLPs are important in model's safety. Our work demostrated MLP in specific inference positions is critical for intruction-tuned model's safety. A more detailed explanation of the differences between our work and previous approaches is provided in the general comment.
>
> > L66 "We evaluate our methods and compare them with other jailbreak approaches". Specify the name explicitly
> >
> > L66, " As a result" result of what?
> >
> > L69, "has a smaller impact on the model" how small?
> >
> > L73, "presents new findings", " Based on these insights", what findings and what insights? can you list them as bullets?
> >
> > L140, suggest to move prompt template details to appendix
> >
> > L222. Drop this sentence "We now complete it here. "
>
> We appreciate the reviewer’s effort in improving the grammar and clarity of our paper. We will improve our writing in the revised paper.

---

> > ### Comment · Reviewer_NJHF · 2024-11-23
> >
> > Thanks for the response.
> >
> > > Previous works have demonstrated MLPs are important in model's safety
> >
> > Needs citation
> >
> > After reading all the other reviews, I decided to keep my rating.

---

### Official Review · Reviewer_jnfo · 2024-11-03

**Soundness:** 2
**Presentation:** 2
**Contribution:** 2
**Rating:** 3
**Confidence:** 3

**Summary:**

The authors investigated the safety mechanisms of instruction-tuned LLMs and discovered a vulnerability through MLP neuron re-weighting. They developed two white-box jailbreak methods: a prompt-specific method that targets individual prompts and a prompt-general method that works on unseen harmful prompts. The study focused specifically on MLP layers in end-of-sentence inferences, finding that modifying these layers was sufficient to compromise model safety. Their prompt-specific method achieved impressive results, with an ASR of 94-97% across various models. The prompt-general method also performed well, achieving an ASR of 67-94%.

**Strengths:**

---> This paper provides two different strategies for white-box jailbreaks in LLMs through MLP neuron reweighting, which not only play a crucial role in understanding and improving the security of LLMs, but also can be used to identify MLP neurons that are strongly correlated with safety.


---> This paper demonstrates the effectiveness of the attack with a high ASR across multiple open-source LLMs, ranging from 2B to 72B parameters.

**Weaknesses:**

-->> 1:   If the dataset used in pre-training consists of harmful questions and responses, then it is likely that the model's objective has been altered to respond harmful questions.

-----> In instruction-tuned LLMs, safety training is a post-process that follows pretraining. Continuing pretraining with harmful datasets will override the safety training.

I think the overall approach is similar to fine-tuning LLMs with adversarial datasets, and I do not believe this proposed approach is novel.

--->> 2: The authors should have performed the experiments using the fine-tuning approach from Qi et al. (2023) and compared the results with the proposed approach.

------>> As this is a white-box attack, the paper should provide a more comprehensive comparison of results with other attacks, as well as with other methods that have been used to create uncensored LLMs, such as Abliteration method (Maxime Labonne, 2024).

---->>>3:  Line 319: “Meanwhile, our method achieves over a 5x improvement in computational speed compared to existing methods.”---> I  could not find any supporting evidence in the paper to back this claim.

---->>> The paper could have been strengthened by including a cost-of-attack comparison with other existing attacks.

**Questions:**

Evaluation by Llama-Guard-3: Guardrail LLMs output is binary, categorizing responses as SAFE and UNSAFE, whereas there may also be cases of neutral responses.

Have the authors conducted further evaluation using other judge LLMs or human evaluation to investigate mixed responses, such as, “Sure, here is how to build <bad-stuff>. However, as an AI, I cannot assist with this.” ?

---

> ### Author Response · Authors · 2024-11-15
>
> > 1: ...
> > I think the overall approach is similar to fine-tuning LLMs with adversarial datasets, and I do not believe this proposed approach is novel.
>
> As explained in the general comment, the key difference between our method and other fine-tuning approaches lies in the restrictions we impose on the modification, which are based on a new phenomenon we discovered. These restrictions provide our method with more robust generalization compared to other fine-tuning methods. If one were directly fine-tuning a model using the GCG loss function, they would quickly find that the model tends to overfit and fails to generate truly harmful responses. Instead, the model would likely produce repeated, non-harmful responses such as "Sure, here is how to Sure, here is how to Sure, here is how to ...".
>
> > 2: The authors should have performed the experiments using the fine-tuning approach from Qi et al. (2023) and compared the results with the proposed approach.
> >
> > As this is a white-box attack, the paper should provide a more comprehensive comparison of results with other attacks, as well as with other methods that have been used to create uncensored LLMs, such as Abliteration method (Maxime Labonne, 2024).
>
> In the revised paper, we will provide comparison between our methods and some fine-tuning approaches. However, we believe it is not entirely fair to compare our method with those that require harmful responses as labels, even though our results are comparable in performance.
>
> > 3: Line 319: “Meanwhile, our method achieves over a 5x improvement in computational speed compared to existing methods.”---> I could not find any supporting evidence in the paper to back this claim.
> >
> > The paper could have been strengthened by including a cost-of-attack comparison with other existing attacks.
>
> We agree that a more comprehensive cost comparison with other existing attacks would strengthen the paper. However, due to space limitation, we were unable to provide a detailed cost-of-attack comparison in the paper. To provide some context, in our experiments with LLaMA-3 8B-Instruct on a single A800 GPU, the GCG method took approximately 20 to 30 minutes to optimize a prompt-specific attack, whereas our prompt-specific method required only 3 to 5 minutes.
>
> > Evaluation by Llama-Guard-3: Guardrail LLMs output is binary, categorizing responses as SAFE and UNSAFE, whereas there may also be cases of neutral responses.
> >
> > Have the authors conducted further evaluation using other judge LLMs or human evaluation to investigate mixed responses, such as, “Sure, here is how to build <bad-stuff>. However, as an AI, I cannot assist with this.” ?
>
> The concerns about the evaluation method are very reasonable, as many rule-based evaluation methods can provide inflated results. In our study, we tested multiple evaluation methods using our jailbreak results for the Llama-3 8B-Instruct model. For a small subset of results, we conducted human evaluation, and for the full set, we used the GPT-4 API to assess the responses. The results from both human evaluation and the GPT-4 API were consistent with those from Llama-Guard-3. As a result, we chose Llama-Guard-3 for our evaluation, as it can be performed locally. We will include the evaluation results from the GPT-4 API in the revised version of the paper.
>
> Regarding the binary output of Llama-Guard-3, we acknowledge that this could be a limitation. However, defining a clear and rigorous "level of harmfulness" for a response is challenging, and we do not fully trust LLMs to handle this task reliably. We welcome further discussion and potential solutions to this problem.

---

> > ### Comment · Reviewer_jnfo · 2024-11-25
> > **Thank you for the response.**
> >
> > I have read through the authors' response as well as the other reviewers' comments. I have decided to keep my rating. I suggest that the authors revise the paper based on the reviewers' comments and submit it to future conferences.

---

### Official Review · Reviewer_yAas · 2024-11-03

**Soundness:** 2
**Presentation:** 2
**Contribution:** 2
**Rating:** 5
**Confidence:** 4

**Summary:**

The paper explores vulnerabilities in instruction-tuned LLMs and introduces new methods to jailbreak these models by re-weighting neurons in multi-layer perceptrons (MLPs) at the end of sentence inferences. The authors propose two jailbreak approaches: a prompt-specific method that targets individual prompts by adjusting MLP weights dynamically, and a prompt-general method that trains on a dataset to bypass safety mechanisms across various prompts. Their findings reveal that end-of-sentence MLP layers play a crucial role in determining whether prompts are harmful, which current safety mechanisms rely upon. Testing on seven popular open-source LLMs, the methods achieve high attack success rates, with the prompt-general method showing generalizability to unseen prompts. The research highlights significant gaps in LLM safety mechanisms, as modifying only specific MLP layers is sufficient to compromise model safety. This work provides insights into LLM vulnerabilities and emphasizes the need for more robust safeguards in future models, while also using MLP re-weighting as a tool for mechanism interpretability to understand which neurons impact safety alignment most.

**Strengths:**

+ They propose two effective jailbreak methods—prompt-specific and prompt-general—achieve high attack success rates across various LLMs.
+ The prompt-general method generalizes well to new harmful prompts, suggesting broader applicability.
+ Introduces an interpretability tool by identifying specific neurons involved in safety, which could aid future model alignment studies.
+ Provides empirical evidence that safety assessments may concentrate during end-of-sentence inferences, offering valuable insights for future safety research.

**Weaknesses:**

+ The method is largely heuristic-driven, meaning the approach lacks rigorous theoretical grounding and may not represent an optimal solution. The loss function, designed to encourage jailbreaks, is based on intuition rather than a systematic analysis.
+ The paper only hypothesizes the role of MLP layers in end-of-sentence safety checks but does not conclusively establish their function in the model’s safety mechanisms, making some conclusions speculative.
+ The study doesn’t deeply explore how these MLP modifications affect other model capabilities or whether they introduce unexpected biases.

**Questions:**

Please see the weaknesses.

---

> ### Author Response · Authors · 2024-11-15
>
> > The method is largely heuristic-driven, meaning the approach lacks rigorous theoretical grounding and may not represent an optimal solution. The loss function, designed to encourage jailbreaks, is based on intuition rather than a systematic analysis.
>
> In our view, our method is more "phenomenon-driven" than "heuristic-driven." As described in the general comment, the motivation for our approach stems from a novel phenomenon we discovered. However, we do acknowledge that the loss function we selected is based on intuition and experience, rather than further understanding.
>
> > The paper only hypothesizes the role of MLP layers in end-of-sentence safety checks but does not conclusively establish their function in the model’s safety mechanisms, making some conclusions speculative.
>
> We have studied the effect of MLP re-weighting at end-of-sentences positions. It seems that those MLP neurons being most heavily re-weighted were contributiing to specific feature direction in later inferences' hidden states, and these contribution is done through the self-atteniton. However, we haven't collect enough evidence to prove these findings. Thus, we choose to state the function of end-of-sentence MLP layers as a hypothesis.
>
> > The study doesn’t deeply explore how these MLP modifications affect other model capabilities or whether they introduce unexpected biases.
>
> We evaluate the impact of our method using standard LLM evaluation benchmarks, as shown in Table 2. These results demonstrate that our modifications do not harm the model's existing capabilities. As for the unexpected bias, while there are no obvious biases introduced by our method, this is still a question that deserves further study.

---

### Official Review · Reviewer_FpVz · 2024-11-04

**Soundness:** 3
**Presentation:** 2
**Contribution:** 2
**Rating:** 3
**Confidence:** 4

**Summary:**

The paper introduces a new method for white-box LLM jailbreak by relocatting the attention map of the last layer of the LLMs (the MLP layer). The paper introduces a new MLP re-weighting method, where, with the re-weighting, the models are more easily to be jailbreaked.

**Strengths:**

1. the idea is illustrated fairly clearly, in particular, I find section 2 and the equation at line 113 helpful, it helps setting up the ground and demosntrating the main focus on this paper is about optimizing for M.

2. Table 2 also illustrates quite clearly.

**Weaknesses:**

1. The biggest concern is that, this paper seems to build upon an unrealistic scenario. The paper assumes that one can have quite transparent access to the LLM (as much as plugging a new component to the existing model) that one desires to jailbreak, this is almost not possible in practice. The only application I can think is that, a highly technical malicious team obtains an open-source model, and then do all the engineering work to jailbreak it for information that an aligned, open-source model is not supposed to easily reveal. This application scenarios seems too specific and ideal in comparison to most of other concurrent jailbreak research that study how a party can jailbreak LLMs without access to weights, gradients, or even with the guardrail functions in fully commercial setting. I would recommend the authors to offer more detailed discussions on the motivation from a practical perspective.

2. The empirical scope of the paper is a bit too humble, it only compares to some popular jailbreak methods and only in HarmBench alone. Considering the whitebox nature of this method, I would recommend the authors to try recently, highly malicious JAMbench, which might show more advantages of the proposed method. If the authors prefer to test only on HarmBench, some additional discussions of the rationale behind it will be preferred or even necessary.

3. The method requires training, so it will be necessary to demonstrate the usability where $M$ is trained over one benchmark (HarmBench, AdvBench, JAMbench, etc) and applied at other benchmarks. This is particularly important if the authors want to claim their methods can compete and be better than concurrent jailbreak methods, as other jailbreak methods does not require training. (More details behind the claim at line 276-278 are necessary). I would recommend the authors to report these results or clarify with more details if such experiments have already been done.


minor:

1. line 093 the double quotes are not renderred correctly.
2. The equation at line 113 might also deserve a number.
3. Related work needs to be updated to include more recently published jailbreak papers. I understand these research are fast developing and it's hard to keep tracks of all the papers online, but even just the ones are officially published are not properly documented here.

**Questions:**

1. Not very sure what the authors mean at line 116, different re-weighting factors for different inferences. It seems to me that this refers to prompt-specific method (3.1) and then contradicts the prompt-general method (3.2). I believe the claim at section 2.1 and line 116 are supposed to serve as the background for the entire paper.

2. What will happen if we set \rho to zero? (Figure 4).

**Details Of Ethics Concerns:**

The research itself is about AI security, thus might require some additional layer of caution, although personally I don't think the authors are crossing lines.

---

> ### Author Response · Authors · 2024-11-15
>
> > Not very sure what the authors mean at line 116, different re-weighting factors for different inferences. It seems to me that this refers to prompt-specific method (3.1) and then contradicts the prompt-general method (3.2). I believe the claim at section 2.1 and line 116 are supposed to serve as the background for the entire paper.
>
> Clarifying this point is essential, as it is indeed the foundation of the entire paper. So, in this paper, we treat decoder-only transformers as sequential models that takes one token at a time. Thus, when we say 'one inference', what we refer to is a forward pass that takes one single token as input. For instance, to read the prompt 'A B C' and generate 'D' as output, the model perform three inferences —- one reading "A", one reading "B", and the last one reading "C" and generating "D". We train independent MLP re-weighting factors for different inferences.
>
> > The biggest concern is that, this paper seems to build upon an unrealistic scenario. The paper assumes that one can have quite transparent access to the LLM (as much as plugging a new component to the existing model) that one desires to jailbreak, this is almost not possible in practice. The only application I can think is that, a highly technical malicious team obtains an open-source model, and then do all the engineering work to jailbreak it for information that an aligned, open-source model is not supposed to easily reveal. This application scenarios seems too specific and ideal in comparison to most of other concurrent jailbreak research that study how a party can jailbreak LLMs without access to weights, gradients, or even with the guardrail functions in fully commercial setting. I would recommend the authors to offer more detailed discussions on the motivation from a practical perspective.
>
> As described in the general comment, the motivation behind our methods arises from our newly discovered phenomenon. We leverage our findings about instruction-tuned LLMs to develop the methods presented in this paper.
>
> Regarding the scenario concern, we acknowledge that the white-box setting is less practicality comparing with black-box approaches. However, there are still meaningful use cases. For instance, one might aim to release an uncensored version of a publicly available open-source model, such as Llama-3 Instruct. While the original instruct model is fine-tuned to enforce safety constraints, our method can generate an uncensored variant using only a few hundred harmful prompts (without requiring any harmful responses).
>
> > The empirical scope of the paper is a bit too humble, it only compares to some popular jailbreak methods and only in HarmBench alone. Considering the whitebox nature of this method, I would recommend the authors to try recently, highly malicious JAMbench, which might show more advantages of the proposed method. If the authors prefer to test only on HarmBench, some additional discussions of the rationale behind it will be preferred or even necessary.
> >
> > The method requires training, so it will be necessary to demonstrate the usability where is trained over one benchmark (HarmBench, AdvBench, JAMbench, etc) and applied at other benchmarks. This is particularly important if the authors want to claim their methods can compete and be better than concurrent jailbreak methods, as other jailbreak methods does not require training. (More details behind the claim at line 276-278 are necessary). I would recommend the authors to report these results or clarify with more details if such experiments have already been done.
>
> In the revised paper, we included additional experimental results on other benchmarks, including AdvBench and JAMBench. See Appendix B (L718-739).
>
> > What will happen if we set \rho to zero? (Figure 4).
>
> If we set $\rho$ to zero, the method will still work. The only difference is that there will be no sparsity requirement for the modification applied to the model. This lack of sparsity weakens the generalization of the results. However, there is no fundamental difference between $\rho = 0$ and a very small $\rho$.

---

> ### Comment · Reviewer_FpVz · 2024-11-25
> **re rebuttal**
>
> Thanks for the clarifications, some of the texts address my concerns, but manuscript does not seem to get updated to the latest versions? for example, Figure 6 in appendix B does not seem to have comparisions of other methods, and Figure 4 still does not have the case  \rho set to zero? Is the last sentence from the rebuttal "there is no fundamental difference ... " just an argument instead of obtained from experiments?

---

### Author Response · Authors · 2024-11-15
**General comment**

Several reviewers have raised questions about the difference between our method and other fine-tuning approaches. In response, we would like to clarify the motivation and novelty of our work.

The motivation for this paper stems from a phenomenon we discovered: applying modifications at a few specific inference positions *alone* can significantly impact model's performance (L193-195). This is novel, since previous works have generally treated inferences at different token positions uniformly. One clear advantage of modifying only a few inferences is the ability to maintain the model's general functionality. A major challenge with fine-tuning and other post-training modifications is their tendency to degrade the model's original capabilities—a problem faced by many fine-tuning-based jailbreak methods. By restricting modifications to a small number of inference token positions (4 out of $n+$), our method preserves the model's original functionality and achieves more robust generalization.

---

### Note · Authors · 2024-11-26

**Comment:**

After reviewing the comments from the reviewers, we believe our work has significant room for improvement. Therefore, we have decided to withdraw and continue refining it. We sincerely thank the reviewers for their efforts.

**Withdrawal Confirmation:**

I have read and agree with the venue's withdrawal policy on behalf of myself and my co-authors.